# Sources of stress, coping strategies and associated factors among Vietnamese first-year medical students

**Tan Nguyen[1], Christy Pu[1], Alexander Waits[1], Tuan D. Tran[2], Yatan Pal Singh Balhara[3], Quynh Thi Vu Huynh[2], Song-Lih Huang[1] ***

**1** Institute of Public Health, National Yang Ming Chiao Tung University, Taipei, Taiwan, **2** University of Medicine and Pharmacy, Ho Chi Minh City, Vietnam, **3** Department of Psychiatry, All India Institute of Medical Sciences (AIIMS), New Delhi, India

* songlih.huang@nycu.edu.tw

## Abstract

### Objectives

This study aims to examine the sources of stress among first-year medical students; the frequency of their coping strategies; the factors associated with specific stressors and specific coping strategies adopted by the participants.

### Methods

We conducted a cross-sectional study with 409 first-year students at the University of Medicine and Pharmacy, Vietnam. The Vietnamese versions of the Higher Education Stress Inventory (V_HESI) and Brief Coping Orientation to Problems Experienced (V_Brief COPE) were validated and were used as measurement instruments for participants' sources of stress and coping strategies frequencies. The survey comprised questions of socioeconomic status, stress-related issues, the six sources of stress (using the V_HESI), and the nine coping strategies (using the V_Brief COPE).

### Results

Among the six sources of stress, "Worries about future competence/endurance" had the highest mean score (3.02±0.64), while "Mismatch in professional role expectations" had the lowest score (1.60±0.53). "Financial concerns" and "Academic workloads" were also significant sources of stress. Regarding coping strategies, Self-distraction was most frequently adopted by the participants (2.80 ± 0.68). Problem-solving (2.72±0.53) and seeking Social support (2.62±0.70) were also common adaptive strategies. Avoidance (1.87±0.55) and substance-use (1.27±0.55) were the least frequent strategies. Students who experienced acute stress event were more likely to have financial concerns compared to others. Substance use was positively associated with stressors from "Mismatch in professional role expectations", "Non-supportive educational environment", "Having physical issues" and "Having part-time job". Self-blame was more frequent among students with "Worries about future competence/endurance", "Financial concerns", and "Academic workload". Male

**Data Availability Statement:** The datasets generated and/or analyzed during the current study are not publicly available due to the data protection policy of University of Medicine and Pharmacy, but

are available from the corresponding author on reasonable request. Data requests may be sent to the Medical Ethics Board of the University of Medicine and Pharmacy (hoidongdaoducdhyd@ump.edu.vn).

**Funding:** The author(s) received no specific funding for this work.

**Competing interests:** The authors have declared that no competing interests exist.

student tended to adopt humor strategy (β = 0.19, p = 0.02), while less likely to utilize religious practices (β = -0.21, p = 0.01).

## Conclusions

Two-thirds of the participants reported moderate to high levels of stress. "Worries about future competence/endurance" was the most concerned stressor, followed by "Academic workload", and "Financial concerns". The first-year medical students reported high frequency of utilization "Self-distraction", "Problem-solving" and "Social support" when confronting stress. The findings may help inform the school management to better support students' well-being.

## Introduction

Stress and coping strategies have increasingly been studied, particularly in the population of medical students [1–5]. Studies in Vietnam show that the prevalence of experiencing stress among medical students are high, with more than 30% of the students perceived moderate to high level of stress [6, 7]. Being exposed to prolonged and high levels of stress may lead to negative consequences: cognitive and emotional burden, psychological distress, dropping out of school, low quality of life and decreased empathy with patients, which all may result in medical errors. These may influence the quality of health care service in the wide aspect [8–10]. Research has suggested that medical students experience stress from different sources, which could be from academic factors, psychosocial factors with emphasis on financial issues, teacher-student relationships, and high expectations from significant others and themselves [11, 12]. In addition, physical issues could also be an important stressor for medical students, which affects their academic performance and quality of life [13, 14]. Furthermore, in medical education settings, previous studies revealed that academic-related stressors may cause more distress than interpersonal, intrapersonal, or environmental stressors [15]. Early identification of these stressors could help tailor appropriate prevention and intervention programs to tackle the problematic psychological issues among this high-risk population [16, 17].

Coping strategies are specific efforts, cognitively and behaviorally, that help a person to tolerate, to minimize or to handle stressors. When confronting stressors, a person may end up with different types of coping strategies depending on the primary appraisal of the stressor as challenging or threatening, then the second appraisal of their available resources to cope with stressors [18]. There're different ways to categorize coping strategies, consisting of problem-focused or emotion-focused, adaptive or maladaptive, approaching or avoiding [19, 20]. Adopting appropriate strategies would help students cope better and healthier with stressors, minimizing the negative consequences from stressors [21–23]. Hence, it is essential to first understand what sources of stress that medical students are facing, then how they are coping with a variety of stressors. Furthermore, the link between potential associated factors with specific coping strategies would contribute to the understanding of coping model, which is important to figure out appropriate approaches to enhance the ability of students to deal effectively with stressors.

To date, few studies have directly examined the associations between specific sources of stress and types of coping strategies, particularly in the early phase of medical training. Our aims are to examine the sources of stress among first-year medical students, the frequencies of

their coping strategies, and to explore the relationships between other factors with specific sources of stress and coping strategies employed by these students.

## Methods

### Study design

We conducted a cross-sectional study among the undergraduate students at the Faculty of Medicine, University of Medicine and Pharmacy, Ho Chi Minh City, Vietnam.

### Participants

In this study, we invited all first-year students. Participants provided written informed consents before participating in the study. Students with informed consents and completed data were included in the study with no explicit exclusion criteria were employed. Amongst 420 students, 409 students participated in this study and completed the questionnaire.

### Procedures

Participants filled in a self-administered survey two months after school admission. The survey was anonymous with no information to identify the participants. Data were collected from 12th November 2020 to 22nd December 2020, and have been accessed for data analysis since 24th December 2020. The principal investigator introduced the purpose of the study to all first-year students in an orientation at the beginning of the academic year. At the end of the orientation, the students sent back the consent forms to members of the research team. We sent a link of the survey to all students with written informed consent. The online survey was designed so all questions need to be completed before submission. The instructions were also given on how to fill in the online questionnaires, which consisted of four sections as mentioned below.

The survey included: (i) Demographic information (Age, Sex, Parental education status, Part-time job); (ii) Self-rated stress level, acute stress event, physical issues, psychological issues, Covid-19-related stress; (iii) Higher Education Stress Inventory (HESI) to assess the sources of stress; (iv) Brief Coping Orientation to Problems Experienced (Brief COPE) to assess the coping strategies. Level of stress was scored from 1 to 3 for low to high level, respectively. Mother and father educational levels were categorized into three levels (Elementary or lower, High school, and College or higher). Part-time job, acute stress event, physical stress, psychological stress, and Covid-19-related stress were binary variables with responses of yes or no.

### Measurement tools

The HESI and Brief COPE went through five steps of validation including: (i) Forward translation, (ii) Backward translation, (iii) Assessment of content validity, (iv) Assessment of factor structure, and (v) Assessment of internal consistency. During the first step, the English instruments were translated from English to Vietnamese separately by two translators (the principal investigator and another team member who has the bachelors in both psychology and English. The second step was conducted by two Vietnamese who has lived in the United state for than 10 years and used English in their daily work. Any ambiguities or discrepancies in terms of context meaning or colloquialism were discussed and resolved through consensus among research team members. The results of the fourth and fifth steps are elaborated as following:

**HESI.**   The original HESI was developed by Dahlin et al. in Sweden among medical students [24]. The strength of this scale is its ability to assess various sources of stress in different higher education settings. From 33 items from the original version, the Exploratory Factor Analysis (EFA) and Confirmatory Factor Analysis (CFA) to get the revised scale with 20 items,

categorized into six sources of stress. Cronbach's alpha was 0.73 for the whole scale. CMIN/DF was 1.97, CFI was 0.901, and RMSEA < 0.06, which indicates adequate degree of model fitting. It can be considered that the above extracted results are feasible and acceptable for this study population. Respondents rate each statement of these 20 items on a 4-point Likert-type scale: 1 (Totally disagree) to 4 (Totally agree).

The six sources of stress included:

1. Mismatch in professional role expectations (from four items "The training demands that I join in situations that I find unethical"; "The professional role presented in the training conflicts with my personal view"; "I feel that I am less well treated because of my ethnic background"; and "I feel that I am less well treated because of my sex").

2. Worries about future competence/endurance (from three items "I worry about long working hours and responsibilities in my future career"; "The insight I have had into my future profession has made me worries about the stressful workload"; "I am worried that I will not acquire all the knowledge needed for my future profession").

3. Financial concerns (from three items "As a student, my financial situation is a worry", "I am worried about my future economy and my ability to repay students loans"; "I am worried about accommodation").

4. Academic workload (from three items "The literature is too difficult and extensive"; "The space of studies is too high"; "Studies control my life and I have little time for other activities").

5. Low identity of medical profession (from three items "I am satisfied with my choice of career"–reversed score; "I am proud of my future profession"–reversed score; "I am able to influence my studies"–reversed score).

6. Non-supportive educational environment (from four items "Student union activities promote a sense of community and contribute to a better working environment for students"–reversed score; "The teachers often give feedback on students' knowledge and skills"–reversed score; "I feel that the training is preparing me well for my future profession"–reversed score; "My fellow students support me"–reversed score)

The score of each source of stress was the average score of their related items, ranging from 1.0 to 4.0.

**BRIEF COPE.** Brief COPE was developed as a short version of the original 60-item COPE scale, by Carver et al (1989) [25], (Carver, 1997) [26]. Brief COPE is a 28-item measure, which is designed to assess the frequency of utilization of various coping strategies on a scale of 1 (I haven't been doing this at all) to 4 (I've been doing this a lot). Our findings from EFA and CFA of the original Brief COPE revealed the revised 27-item scale with 9 categories of coping strategies. Cronbach's alpha was 0.78. CMIN/DF is 1.99, CFI is greater than 0.9, RMSEA < 0.06, and SRMR < 0.08 which indicates a good degree of model fitting, and the above extracted results are feasible and acceptable for this study population.

The nine coping strategies included:

1. Problem solving (from seven items "I've been taking action to try to make the situation better"; "I've been concentrating my efforts on doing something about the situation I'm in"; "I've been trying to come up with a strategy about what to do"; "I've been thinking hard about what steps to take"; "I've looking for something good in what is happening"; "I've been learning to live with it"; "I've been trying to see it in a different light, to make it seem more positive").

2. Social support (from four items "I've been getting help and advice from other people"; "I've been getting emotional support from others"; "I've been getting comfort and understanding from someone"; "I've been trying to get advice or help from other people about what to do").

3. Avoidance (from four items "I've been giving up the attempt to cope"; "I've been refusing to believe that it has happened"; "I've been giving up trying to deal with it"; "I've been saying to myself 'this isn't real'").

4. Substance use (from two items "I've been using alcohol or other drugs to make myself feel better"; "I've been using alcohol or other drugs to help me get through it").

5. Self-blame (from two items "I've been criticizing myself"; "I've been blaming myself for things that happened").

6. Religion (from two items "I've been praying or meditating"; "I've been trying to find comfort in my religion or spiritual beliefs")

7. Humor (from two items "I've been making fun of the situation"; "I've been making jokes about it").

8. Venting (from two items "I've been saying things to let my unpleasant feeling escape"; "I've been expressing my negative feelings").

9. Self-distraction (from two items "I've been doing something to think about it less, such as going to movies, watching TV, reading, daydreaming, sleeping or shopping"; "I've been turning to work or other activities to take my mind off things").

The score of each coping strategy was the average score of their related items, ranging from 1.0 to 4.0.

## Analytical approach

We used RStudio [27] and SPSS 26 for data analyses. Descriptive statistics (mean, standard deviation, and percentage) were used for calculating the frequencies and proportions of demographic variables, sources of stress, and coping strategies. To assess the factors associated with sources of stress and coping strategies, multiple linear regressions, with forced entry regressions, were estimated. For each source of stress as dependent variable, the independent variables included all sociodemographic factors: sex, age, parental educational levels, part-time job, level of perceived stress, and other stress-related variables (acute stress event, physical stress, psychological stress, Covid-19-related stress). Similarly for each coping strategy as dependent variable, we run different regression models that included different sources of stress and all the above sociodemographic and stress-related factors as independent variables.

This study was approved by the IRB of National Yang Ming Chiao Tung University and University of Medicine and Pharmacy, and guidelines for research with human subjects were followed.

## Results

### Sociodemographic of participants

Table 1 shows the characteristics of the study population. Among the 409 participants, the mean age was 18±0.38 years. Approximately 40% of participants were female. Most of the participants' parents finished their high school or higher education. More than 20% of students reported experiencing either physical issues or psychological issues, while less than 10% had

**Table 1. Characteristics of the study participants.**

|  | N = 409 students |
| --- | --- |
| Age |  |
| Mean (SD) | 18.1 (0.38) |
| Sex |  |
| Female (N, %) | 151 (36.9%) |
| Male (N,%) | 258 (63.1%) |
| Father education |  |
| Primary school or lower (N, %) | 11 (2.7%) |
| High school (N, %) | 108 (26.4%) |
| College or higher (N, %) | 290 (70.9%) |
| Mother education |  |
| Primary school or lower (N, %) | 21 (5.1%) |
| High school (N, %) | 161 (39.4%) |
| College or higher (N, %) | 227 (55.5%) |
| Part-time job |  |
| Yes (N, %) | 32 (7.8%) |
| No (N, %) | 377 (92.2%) |
| Acute stress |  |
| Yes (N, %) | 119 (29.1%) |
| No (N%) | 290 (70.9%) |
| Physical stress |  |
| Yes (N, %) | 98 (24.0%) |
| No (N, %) | 311 (76.0%) |
| Psychosocial stress |  |
| Yes (N, %) | 93 (22.7%) |
| No (N, %) | 316 (77.3%) |
| Covid-19—related stress |  |
| Yes (N, %) | 39 (9.5%) |
| No (N, %) | 370 (90.5%) |
| Stress level |  |
| Low (N, %) | 142 (34.7%) |
| Moderate (N, %) | 231 (56.5%) |
| High (N, %) | 36 (8.8%) |

stress due to Covid-19. Among the participants, there were 231 students (57%) reported a moderate level of stress, and 36 students (8.9%) had a high level of stress (Table 1).

## The sources of stress and the coping strategies adopted by the participants

Fig 1 illustrates the sources of stress among the study participants, using the Vietnamese version of HESI scale. The HESI subscale scores are listed in descending order from the participants. "Worries about future competence/endurance" had the highest mean score among 409 participants (3.02±0.64), while "Mismatch in professional role expectations" had the lowest score (1.60±0.53). "Financial concerns" and "Academic workloads" were also significant sources of stress among first-year students in this study, with the mean scores of 2.65 and 1.86, respectively. The other stressors with less concerns were "Low identity of medical profession" (mean score: 1.87) and "Non-supportive educational environment" (mean score: 1.80) (Fig 1).

The coping strategies scores of the participants are shown in Fig 2. Regarding coping strategies employed by the study population, Self-distraction was the most frequently reported

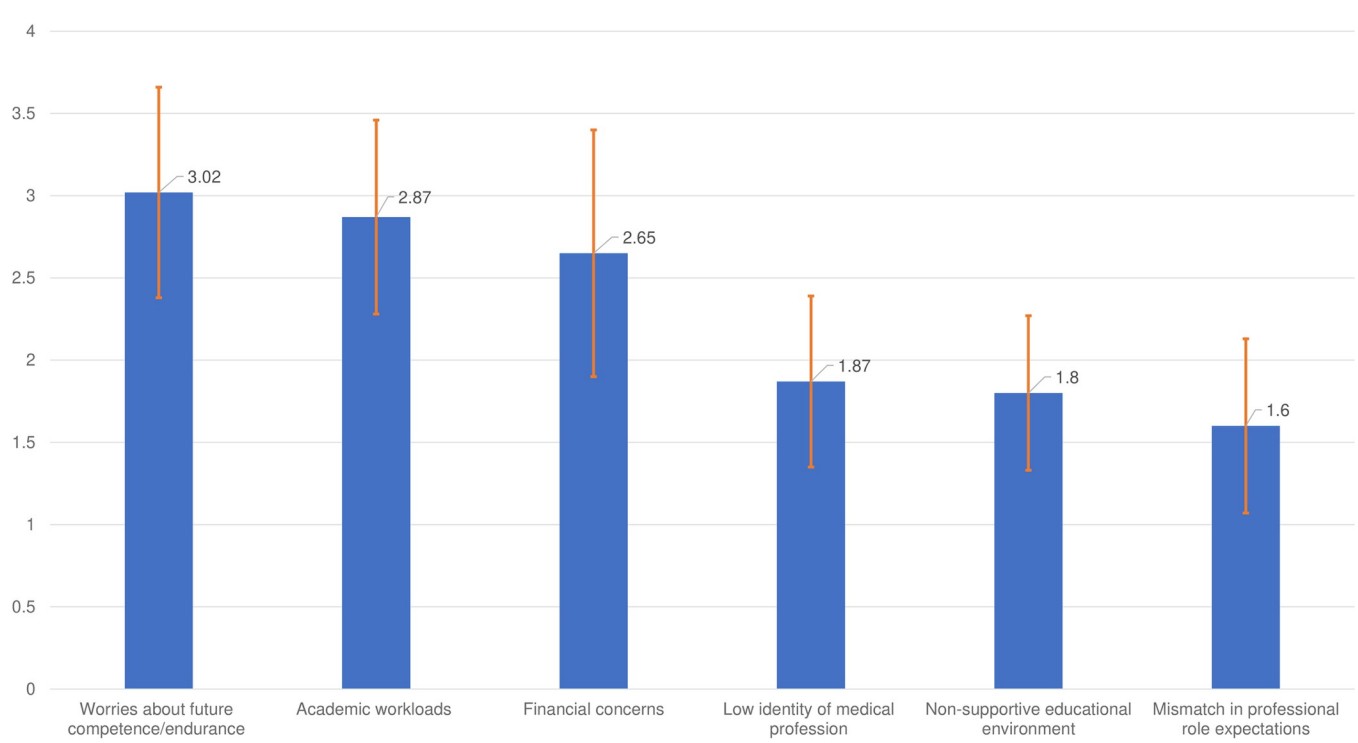

**Fig 1. Sources of stress among 409 students, using the revised 20-item HESI scale.**

among these 409 students (2.80 ± 0.68). However, Problem-solving and seeking Social support were predominant the other less adaptive strategies, with the scores were 2.72 (0.53) and 2.62 (0.70), respectively. Self-blame also had a high score (2.52±0.74), i.e., it appears quite frequently among the participants. Avoidance (1.87±0.55) and substance-use (1.27±0.55) were the least frequent strategies adopted to cope with stress (Fig 2).

## Factors associated with sources of stress and coping strategies adopted by the participants

Table 2 shows the factors that are associated with specific source of stress among the participants. Students who reported high perceived stress level are more likely to get stressed due to "Mismatch in professional role expectations", "Worries about future competence/endurance", and "Non-supportive educational environment". Students experiencing acute stress event were more likely to have financial concerns compared to other students (β = 0.20). On the other hand, "Low identity of medical profession" was found associated with female students and students who had part-time jobs. Female students are also more likely to have "Worries about future competence/endurance".

Table 3 illustrates a variety of associated factors with coping strategies utilized by the participants. Students with "Low identity of medical profession" were less likely to have orientations to solve the problem (β = -0.11) or to seek social support (β = -0.25). Meanwhile, students with stressors due to "Mismatch in professional role expectations" tend to have both adaptive and maladaptive strategies: Humor (β = 0.25), Religion (β = 0.3), Avoidance (β = 0.22), and Substance use (β = 0.25). Substance use was also found positively associated with stressors from "Non-supportive educational environment", "Having physical issues" and "Having part-time jobs". On the other hand, Self-blame was reported higher frequency among students with

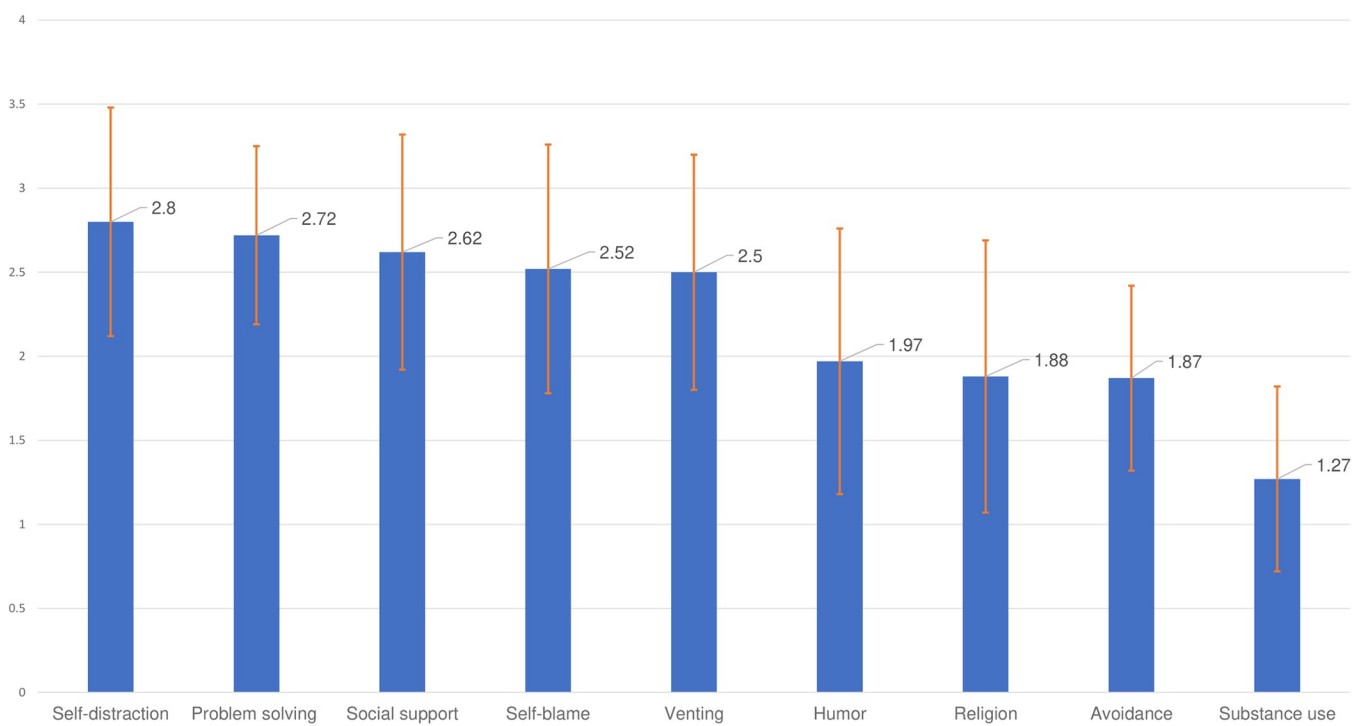

**Fig 2. Coping strategies scores of 409 students, using the revised 27-item brief COPE scale.**

"Worries about future competence/endurance" ($\beta = 0.14$), "Financial concerns" ($\beta = 0.17$), "Academic workload" ($\beta = 0.18$). Male student tent to adopt humor strategy ($\beta = 0.19$, $p = 0.020$), while less likely to utilize religious practices ($\beta = -0.21$, $p = 0.013$). Meanwhile, students experiencing psychological issues tend to not use "Self-distraction" as a coping orientation to their problems ($\beta = -0.23$, $p = 0.013$).

**Table 2. Factors associated with the six sources of stress.**

| Factors | Estimates | p-value |
|---|---|---|
| **Mismatch in professional role expectations** | | |
| Low perceived stress level | -0.32 | 0.003 |
| **Worries about future competence/endurance** | | |
| Being male student | -0.23 | <0.001 |
| Low perceived stress level | -0.28 | 0.033 |
| **Financial concerns** | | |
| Experiencing acute stress event | 0.20 | 0.027 |
| **Low identity of medical profession** | | |
| Being male student | -0.14 | 0.007 |
| Having part-time job | 0.28 | 0.005 |
| **Non-supportive educational environment** | | |
| Low perceived stress level | -0.28 | 0.004 |
| Moderate perceived stress level | -0.19 | 0.030 |

For each source of stress, we ran a separate regression. The regression models were adjusted for: sex, age, parental educational levels, part-time job, level of perceived stress, acute stress event, physical stress, psychological stress, Covid-19-related stress.
Only factors with statistical significance were listed in Table 2.

**Table 3. Factors associated with the nine coping strategies.**

| Factors | Estimates | p-value |
|---|---|---|
| **Problem solving** | | |
| Low identity of medical profession | -0.11 | 0.029 |
| **Social support** | | |
| Low identity of medical profession | -0.25 | <0.001 |
| **Avoidance** | | |
| Mismatch in professional role expectations | 0.22 | <0.001 |
| **Substance use** | | |
| Mismatch in professional role expectations | 0.25 | <0.001 |
| Non-supportive educational environment | 0.15 | 0.011 |
| Having physical issues | 0.15 | 0.033 |
| Having part-time job | 0.20 | 0.048 |
| **Self-blame** | | |
| Worries about future competence/ endurance | 0.14 | 0.013 |
| Financial concerns | 0.17 | <0.001 |
| Academic workload | 0.18 | 0.003 |
| **Religion** | | |
| Mismatch in professional role expectations | 0.30 | <0.001 |
| Being male student | -0.21 | 0.013 |
| **Humor** | | |
| Mismatch in professional role expectations | 0.25 | <0.001 |
| Being male student | 0.19 | 0.020 |
| **Self-distraction** | | |
| Having psychological issues | -0.23 | 0.013 |

For each coping strategy, we ran a separate regression. The regression models were adjusted for: sources of stress, sex, age, parental educational levels, part-time job, level of perceived stress, acute stress event, physical stress, psychological stress, Covid-19-related stress.

Only factors with statistical significance were listed in Table 3.

## Discussions

This study examined the sources of stress and the coping strategies adopted by first-year medical students at University of Medicine and Pharmacy at Ho Chi Minh City, Vietnam. Analyses were conducted to investigate whether there are associations between specific sources of stress and specific coping strategies. We found that specific sources of stressors were linked to specific strategies when students confront stress. Either "Worries about future competence/ endurance", "Financial concerns" or "Academic workload" was found positively associated with Self-blame. On the other hand, when students had "Low identity of medical profession", they were less likely to solve the problem or to seek social support (emotional or instrumental support). While "Mismatch in professional role expectations" was positively associated with the likelihood of adoption different coping strategies, namely Humor, Religion (Religious practice), Avoidance, and Substance use strategies; the source of stress named "Non-supportive educational environment" was only significantly associated with Substance use among the participants. Moreover, students with physical issues were more likely to use substances to deal with stress compared to those reported no stress due to physical issues. These associations were found independent of demographic factors, psychosocial factors and stress-related factors.

The results reveal that two-thirds of the participants rated their stress levels were moderate to high level after admission two months. This is comparable to other studies from Vietnam and other medical school in other countries [28, 29]. A cross-sectional study conducted among 411 first-year students at the University of Medicine of Pharmacy of Ho Chi Minh City and Can Tho in 2020 showed almost 50% of students have problems with stress [28]. Another cross-sectional study in 2003 among 686 students at the Faculty of Medicine, at Ramathibodi Hospital in Thailand found that 61.4% of the medical students reported different perceived levels of stress. However, only 2.4% of students rated their stress level as high in that study, whereas our finding showed 8.9% of the participants had high level of stress. Also in the above study, when rating the sources of stress from a questionnaire, the most prevalent sources of stress of these 686 medical students were from academic performance, including Test/exam (99% of the students), Falling behind in reading schedule (~97% of the students), Getting poor marks (~93% of the students), or Heavy workload (85.2% of the students). "Feeling of incompetence" was reported from 78.4%, while poor motivation to learn met in 48% of the students [29]. The sources of stress reported in our studies were measured by the revised 20-items HESI scale, which showed that Worries about future incompetence/endurance was the highest ranked stressor of the participants. "Academic workload" came as the second significant source of stress, followed by "Financial concerns". One remarkable observation was that the scores of the remaining three sources of stress were much lower than the first ranked three stressors above. Previous studies also suggested that psychosomatic discomfort, e.g., menstrual disturbance or polycystic ovary syndrome in female medical students, had negative effects on students' academic performance and their quality of life [13, 14]. This suggests that medical undergraduate students face a variety of stressors, which are not just limited to academic burden, but encompass their self-expectations and motivations to practice medical profession in the future. However, in this study we utilized the HESI instrument that could not examine the physiological stressors of medical students and the associations of these stressors with coping strategies adopted by medical students.

Mismatch in professional role expectations with their inner conflicts to educational activities, served as other sources of stress. Pursuing medical practice requires students to understand patients' suffering and be willing to provide necessary treatment with professionalism and empathy. Accordingly, training activities would build up professional roles with specific expectations, including but not limited to being authentic, ethical, and respectful. Without the necessary supportive system, students would feel frustrated and stressed, particularly in their first year entering medical education. In our study, "Non-supportive educational environment" includes both the school environment and teacher-student relationships. Previous studies focus on students' experiences of homesick, difficulties in adapting new environments, as well as lacks of contact with family [30]; or competitive, cold and impersonal attitudes [31]. Since HESI was not designed specifically for medical education settings, we do not measure other sources of stress from medical training: facing illness or death of patients, parental wish for you to study medicine [15].

The role of gender when investigating stressors in medical students has long been studied, with inconsistent findings. In our study, female students were more likely to get stress due to their worries of future competence/endurance and their low identity of medical profession, but not other sources of stress. This finding was aligned with other studies as non-male gender has been identify as risk factor of medical students' distress [32], particular stressor due to academic workload [33, 34]. On the other hand, study of Yogesh et al of 100 first-year medical students showed that stress levels due to academic and interpersonal issues were lower in female compared to their male counterpart [11]. Another study of Sadiq et al revealed higher levels of all stressors among female students, but no significant correlations between gender and

sources of stress was found [15]. Furthermore, our findings showed that students experiencing acute stress event tend to be stressed due to financial concerns, while students with part-time jobs were more likely to report stress of "Low identity of medical profession". We have not found any evidence of demographic and social factors associated with stressors among medical students in the available research. Understanding the association of these socioeconomic factors and different sources of stress would help school administrators analyze and figure out appropriate approaches to support these students. Further studies may need to explore more these relationships.

Regarding coping strategies, Self-distraction was the most frequently used among 409 participants. However, active coping strategies (Problem-solving, seeking for Social support, Venting) were also popular in this study population. This was aligned with findings of other studies on coping strategies of medical students [35–37]. On the other hand, Humor and Religion (Religious practice) were reported with lower frequency. Religious practicing was one of the most adopted coping strategies in some countries [38, 39], but not in our study population. Positive religious practice has been proved to have positive effect on students' resilience and their mental health [40, 41]. Among the maladaptive coping strategies, Self-blame had a relatively high score, whereas Substance-use and Avoidance were found not frequently adopted by the participants. These are positive findings among the first-year students since previous studies conducted in the United Kingdom and in Nepal found "Substance use" as a very common coping strategies among medical students [42–44]. However, under-reporting cannot be ruled out. Since substance-use was regarded as moral issue in Vietnam society, students might not be open about it despite the anonymity and confidentiality nature of our surveys.

Associations between different factors and coping strategies were observed. Overall, the strength of associations was low to moderate, yet in the expected directions. This study found that the levels of perceived stress was not linked to any specific coping strategies, which was discrepant with the previous findings that high level of stress would increase the use of maladaptive coping strategies including self-blame, substance use, denial, wishful thinking, behavioral disengagement [45, 46]. Male students were more likely to adopt humor strategy than their female counterpart, while less likely to have religious practice. This was similar to the results of a study conducted in a sample of 94 medical students in their third year about the use of humor strategy [47]. This also added new insights into the previous studies about coping strategies' differences among male and female medical students. Though the common findings were that female students tent to adopt emotional and instrumental support, venting and self-distraction [47, 48], we did not find the significant associations between students' gender with these coping strategies. Provided that these associations between stressors and coping strategies have not been studied in the field of medical training, our findings were novel and would shed the light on approaches to enhance students' coping strategies to deal with different sources of stress.

The study has some limitations, including the nature of cross-sectional study could not draw any causal relationship between the sources of stressors and coping strategies adopted by the participants. Also, when using the Brief COPE to measure the frequency of coping strategies adopted by medical students, the items did not specify the sources of stress that result in such coping strategies. Hence, we could not imply the high frequency of any coping strategy due to any specific source of stressors. However, the strength of this study was the participation of 97% of first-year students in a school, enhancing the representative of the first-year medical students at Faculty of Medicine, at University of Medicine and Pharmacy. Furthermore, the instruments used went through the five-step validation among first-year medical students, ensuring the validity and reliability of the scales.

## Conclusions

Medical students were exposed to a variety of stressors since their first year of training, and the results showed that two-thirds of students reporting moderate to high level of stress. "Worries about future competence/endurance" was the most concerned stressor, followed by "Academic workload", and "Financial concerns". The participants reported high frequency of utilization "Self-distraction", "Problem-solving" and seeking "Social support" when dealing with stress. The findings revealed significant associated factors of sources of stress and coping strategies adopted by first-year medical students. These observations may help inform the school administrators and leaders of the status of stressors and coping as well as suggest appropriate approaches to support their students' well-being.

## Supporting information

**S1 Checklist. STROBE statement—checklist of items that should be included in reports of observational studies.**
(DOCX)

## Acknowledgments

We would like to thank our participants for their willingness to complete the surveys. We also would like to express our gratitude to the school leaders for being supportive for us to accomplish this study.

## Author Contributions

**Conceptualization:** Tan Nguyen, Christy Pu, Alexander Waits, Tuan D. Tran, Song-Lih Huang.

**Data curation:** Tan Nguyen, Christy Pu, Alexander Waits, Yatan Pal Singh Balhara.

**Formal analysis:** Tan Nguyen.

**Investigation:** Tan Nguyen.

**Methodology:** Tan Nguyen, Christy Pu, Alexander Waits, Yatan Pal Singh Balhara, Quynh Thi Vu Huynh, Song-Lih Huang.

**Project administration:** Tan Nguyen.

**Software:** Tan Nguyen.

**Supervision:** Christy Pu, Tuan D. Tran, Song-Lih Huang.

**Validation:** Tan Nguyen, Christy Pu, Quynh Thi Vu Huynh, Song-Lih Huang.

**Writing – original draft:** Tan Nguyen.

**Writing – review & editing:** Christy Pu, Alexander Waits, Tuan D. Tran, Yatan Pal Singh Balhara, Quynh Thi Vu Huynh, Song-Lih Huang.

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
