## [Decision Letter · Decision Letter 0]

24 Jul 2023

PONE-D-23-15195Sources of stress, coping strategies and associated factors among Vietnamese first-year medical studentsPLOS ONE

Dear Dr. Huang,

Thank you for submitting your manuscript to PLOS ONE. After careful consideration, we feel that it has merit but does not fully meet PLOS ONE’s publication criteria as it currently stands. Therefore, we invite you to submit a revised version of the manuscript that addresses the points raised during the review process.

We look forward to receiving your revised manuscript.

Kind regards,

Diego A. Forero, MD; PhD

Academic Editor

PLOS ONE

Reviewers' comments:

Reviewer's Responses to Questions

**Comments to the Author**

1. Is the manuscript technically sound, and do the data support the conclusions?

Reviewer #1: Yes

Reviewer #2: Partly

2. Has the statistical analysis been performed appropriately and rigorously? 

Reviewer #1: Yes

Reviewer #2: Yes

3. Have the authors made all data underlying the findings in their manuscript fully available?

Reviewer #1: Yes

Reviewer #2: No

4. Is the manuscript presented in an intelligible fashion and written in standard English?

Reviewer #1: Yes

Reviewer #2: No

5. Review Comments to the Author

Reviewer #1: The authors provided an extensive characterization of stressors and the coping strategies among medical students. The manuscript is well written however, please find my suggestions below. I will appreciate if the authors could address these issues.

1. I understand the sources of external stressors which cause difficulty of quality of life index for medical students. However, it's equally important to indicate the physiological stressors as well. Two recent articles demonstrated how a physiological abnormality like PMS/PMDD may lead to low academic and social life of medical students leading to additional amount of internal stressors. It will be beneficial for our reader and the policy makers at least to be aware of this too. Here are the two links:

DOI: 10.7759/cureus.40141

DOI: 10.3389/fmed.2022.821908

2. Results: Overall comments: The results are expressed in table format which is fine for me however, please use graphical format for better presentation and use the table as supplemental as required. My intention is not to make you work more but to make your manuscript more readable.

Table 1:Characteristics of the study participants. This table can be easily shown with a bar diagram/scatter plot or by any visual graphical way. Therefore, represent the data of table 1 with a graph and attach the table as supplemental if necessary.

Table 2: Sources of stress among 409 students, using the revised 20-item HESI scale. This table has the mean and SD values. Therefore, please use a graph rather than the table to represent the data.

Table 3: Coping strategies scores of 409 students, using the revised 27-item Brief COPE scale. Please do the same for this table as well.

Table 4: Factors associated with the six sources of stress. This table may stay as it is.

Table 5: Factors associated with the nine coping strategies. As per this table some of the factors such as social support has p value of <0.001 whereas the other has >0.001. Therefore, use the data of this table to convert to a graphical chart(may be a bar diagram) and then indicate which bars are significant by adding * on top that bar. I hope this makes sense. This is the traditional way of representing data. Off course, you can add the table as supplemental if required.

Discussion: The discussion is well written however please find my suggestions below.

Generally, the discussion starts with a paragraph stating the problem, hypothesis. However, the authors directly started discussion the result. This disrupts the flow and readability. Please modify.

The discussion section should include references of systematic review and meta-analysis of medical student and stressor. You may consider using these recent references

DOI: 10.7759/cureus.40141

DOI: 10.3389/fmed.2022.821908

Reviewer #2: I could not agree less that the study is germane and would benefit Vietnamese readers. However, several incorrect claims were made in the manuscript. This manuscript needs to be worked on since the journal does not offer copyediting services. Please refer to the PDF file for comments and recommendations.

6. PLOS authors have the option to publish the peer review history of their article (what does this mean?). If published, this will include your full peer review and any attached files.

Reviewer #1: No

Reviewer #2: No

---

## [Author Response · Author response to Decision Letter 0]

19 Feb 2024

Reviewers’ comments

Dear Reviewers, 

We are grateful for your time and effort reviewing our manuscript. Your comments and suggestions directed us to improve our analysis and writing. Kindly refer to the point-by-point revisions we conducted in response to your comments.

Reviewer 1

Dear reviewer, we are very grateful for your time dedicated to reading our manuscript. Your comments provided multiple insights and helped us to improve our writing. We tried our best to carefully address your comments. Kindly refer to the details below. 

1. I understand the sources of external stressors which cause difficulty of quality of life index for medical students. However, it's equally important to indicate the physiological stressors as well. Two recent articles demonstrated how a physiological abnormality like PMS/PMDD may lead to low academic and social life of medical students leading to additional amount of internal stressors. It will be beneficial for our reader and the policy makers at least to be aware of this too. Here are the two links:

DOI: 10.7759/cureus.40141

DOI: 10.3389/fmed.2022.821908

We added the statement and the discussion of the effects of physical issues or physiological stressors on medical students’ academic performance and quality of life, using the references from your recommendation (Line 55-57, Line 300-302, and Line 305-307 in the revised manuscript) in the Introduction section and in the Discussion section, as following: 

- In Introduction section:

“In addition, physical issues could also be an important stressor for medical students, which affects their academic performance and quality of life.”

- In Discussion section:

“Previous studies also suggested that psychosomatic discomfort, e.g., menstrual disturbance or polycystic ovary syndrome in female medical students, had negative effects on students’ academic performance and their quality of life.”

“However, in this study we utilized the HESI instrument that could not examine the physiological stressors of medical students and the associations of these stressors with coping strategies adopted by medical students.”

2. Results: Overall comments: The results are expressed in table format which is fine for me however, please use graphical format for better presentation and use the table as supplemental as required. My intention is not to make you work more but to make your manuscript more readable.

Table 1: Characteristics of the study participants. This table can be easily shown with a bar diagram/scatter plot or by any visual graphical way. Therefore, represent the data of table 1 with a graph and attach the table as supplemental if necessary.

Table 2: Sources of stress among 409 students, using the revised 20-item HESI scale. This table has the mean and SD values. Therefore, please use a graph rather than the table to represent the data.

Table 3: Coping strategies scores of 409 students, using the revised 27-item Brief COPE scale. Please do the same for this table as well.

Table 4: Factors associated with the six sources of stress. This table may stay as it is.

Table 5: Factors associated with the nine coping strategies. As per this table some of the factors such as social support has p value of <0.001 whereas the other has >0.001. Therefore, use the data of this table to convert to a graphical chart(may be a bar diagram) and then indicate which bars are significant by adding * on top that bar. I hope this makes sense. This is the traditional way of representing data. Off course, you can add the table as supplemental if required.

Thank you for your thorough suggestions and instructions on the use of figures in our manuscript. We have added Figure 1 and Figure 2 to replace Table 2 and Table 3 in the old manuscript. For Table 1, however, we’d like to keep the table format to show detailed information about demographic factors of the participants. Similarly, for Table 5, we’d like to keep as it is to have the consistency of data presentation from the factors associated with stressors and with coping strategies.

3. Discussion: The discussion is well written however please find my suggestions below.

Generally, the discussion starts with a paragraph stating the problem, hypothesis. However, the authors directly started discussion the result. This disrupts the flow and readability. Please modify.

We added a paragraph in the Discussion section to state the objectives and our approaches (Line 264-279 in the revised manuscript), as following:

“This study examined the sources of stress and the coping strategies adopted by first-year medical students at University of Medicine and Pharmacy at Ho Chi Minh City, Vietnam. Analyses were conducted to investigate whether there are associations between specific sources of stress and specific coping strategies. We found that specific sources of stressors were linked to specific strategies when students confront stress. Either “Worries about future competence/endurance”, “Financial concerns” or “Academic workload” was found positively associated with Self-blame. On the other hand, when students had “Low identity of medical profession”, they were less likely to seek social support (emotional or instrumental support). While “Mismatch in professional role expectations” was positively associated with the likelihood of adoption different coping strategies, namely Humor, Religion (Religious practice), Avoidance, and Substance use strategies; the source of stress named “Non-supportive educational environment” was only significantly associated with Substance use among the participants. Moreover, students with physical issues were more likely to use substances to deal with stress compared to those reported no stress due to physical issues. These associations were found independent of demographic factors, psychosocial factors and stress-related factors.”

The discussion section should include references of systematic review and meta-analysis of medical student and stressor. You may consider using these recent references

DOI: 10.7759/cureus.40141

DOI: 10.3389/fmed.2022.821908

We added two sentences in the Discussion section to mention the effects of physiological issues on medical students’ academic performance and quality of life as you suggested, and why we did not measure the physiological stressors of medical students (Line 300-302, and Line 305-307 in the revised manuscript), as following:

“Previous studies also suggested that psychosomatic discomfort, e.g., menstrual disturbance or polycystic ovary syndrome in female medical students, had negative effects on students’ academic performance and their quality of life.”

“However, in this study we utilized the HESI instrument that could not examine the physiological stressors of medical students and the associations of these stressors with coping strategies adopted by medical students.”

Reviewer 2

Dear reviewer, we are very grateful for your time dedicated to reading our manuscript. Your comments provided multiple insights and helped us to improve our writing. We tried our best to carefully address your comments. Kindly refer to the details that we’ve revised based on your comments and suggestions from the PDF file.

In summary, there’re some points that we’ve revised to elaborate more details:

- The objectives of this study include examining the associations between specific sources of stress and specific coping strategies. Although there have been multiple studies on the associations between demographic factors and psychosocial factors with sources of stress or with coping strategies. To the best of our knowledge, the associations between these two main outcomes have not been investigated. We revised the statement to be clearer (Line 79-81 and Line 226-229 in the revised manuscript), as following:

“To date, few studies have directly examined the associations between specific sources of stress and types of coping strategies.”

- We added a brief description of the university and the medical program as you suggested (Line 76-79 in the revised manuscript), as following:

“The University of Medicine and Pharmacy (UMP) in Ho Chi Minh City has adopted a reformed medical program, which is outcome-based medical education. This reformed program was considered with higher demands and more stressful than the old program, particularly for first-year students who have just entered medical training.”

- We elaborated our approaches to collect data (Line 93 and Line 99-105 in the revised manuscript), as following:

“Amongst 420 students, 409 students participated in this study and completed the questionnaire.”

“The principal investigator introduced the purpose of the study to all first-year students in an orientation at the beginning of the academic year. At the end of the orientation, the students sent back the consent forms to members of the research team. We sent a link of the survey to all students with written informed consent. The online survey was designed so all questions need to be completed before submission. The instructions were also given on how to fill in the online questionnaires,”

- We moved the variable definitions from the Analytical approach section to Procedure section as you suggested, (Line 110-114 in the revised manuscript), as following:

“Level of stress was scored from 1 to 3 for low to high level, respectively. Mother and father educational levels were categorized into three levels (Elementary or lower, High school, and College or higher). Part-time job, acute stress event, physical stress, psychological stress, and Covid-19-related stress were binary variables with responses of yes or no.”

- We moved the description of instruments’ validation steps from the Analytical approach section to Measurement tools section as you suggested. We also add elaboration on how the items of the questionnaires were modified throughout five-step validation process (Line 116-125 in the revised manuscript), as following:

“The HESI and Brief COPE went through five steps of validation including: (i) Forward translation, (ii) Backward translation, (iii) Assessment of content validity, (iv) Assessment of factor structure, and (v) Assessment of internal consistency. During the first step, the English instruments were translated from English to Vietnamese separately by two translators (the principal investigator and another team member who has the bachelors in both psychology and English. The second step was conducted by two Vietnamese who has lived in the United state for than 10 years and used English in their daily work. Any ambiguities or discrepancies in terms of context meaning or colloquialism were discussed and resolved through consensus among research team members. The results of the fourth and fifth steps are elaborated as following:”

- We added the statement that the survey was designed to have all the items completed before submission, so we did not have missing data from our dataset (Line 103-104 in the revised manuscript), as following:

“The online survey was designed so all questions need to be completed before submission.”

- We revised Table 1 for the wrong numbers in the table. This was because we mistakenly keyed in from the outputs of statistical software. Thank you very much for your revisions.

- We added references, elaborated the references, and revised the writing in the Discussion part regarding the prevalence of stress among medical students (Line 282-294 in the revised manuscript), as following:

“A cross-sectional study conducted among 411 first-year students at the University of Medicine of Pharmacy of Ho Chi Minh City and Can Tho in 2020 showed almost 50% of students have problems with stress. Another cross-sectional study in 2003 among 686 students at the Faculty of Medicine, at Ramathibodi Hospital in Thailand found that 61.4% of the medical students reported different perceived levels of stress. However, only 2.4% of students rated their stress level as high in that study, whereas our finding showed 8.9% of the participants had high level of stress. Also in the above study, when rating the sources of stress from a questionnaire, the most prevalent sources of stress of these 686 medical students were from academic performance, including Test/exam (99% of the students), Falling behind in reading schedule (~97% of the students), Getting poor marks (~93% of the students), or Heavy workload (85.2% of the students). “Feeling of incompetence” was reported from 78.4%, while poor motivation to learn met in 48% of the students”

- We added references, and revised the writing in the Discussion part regarding then religious practice of medical students (Line 343-345 in the revised manuscript), as following:

“Religious practicing was one of the most adopted coping strategies in some countries (32,33), but not in our study population. Positive religious practice has been proved to have positive effect on students’ resilience and their mental health.”

- Besides the above, we’ve revised the typo, rewrote the manuscript, added details in the revised manuscript, along with your comments and suggestions, as you may find:

o Line 272-276 in the revised manuscript, as following:

“While “Mismatch in professional role expectations” was positively associated with the likelihood of adoption different coping strategies, namely Humor, Religion (Religious practice), Avoidance, and Substance use strategies; the source of stress named “Non-supportive educational environment” was only significantly associated with Substance use among the participants.”

o Line 372-374 in the revised manuscript, as following:

“Also, when using the Brief COPE to measure the frequency of coping strategies adopted by medical students, the items did not specify the sources of stress that result in such coping strategies.”

o Line 375-380 in the revised manuscript, as following:

“However, the strength of this study was the participation of 97% of first-year students in a school, enhancing the representative of the first-year medical students at Faculty of Medicine, at University of Medicine and Pharmacy. Furthermore, the instruments used went through the five-step validation among first-year medical students, ensuring the validity and reliability of the scales.”

o Line 382-384 in the revised manuscript), as following:

“Medical students were exposed to a variety of stressors since their first year of training, and the results showed that two-thirds of students reporting moderate to high level of stress.”

- We’ve checked the Reference to make sure it follows the format of the journal, which is Vancouver style.

---

## [Decision Letter · Decision Letter 1]

19 Mar 2024

PONE-D-23-15195R1Sources of stress, coping strategies and associated factors among Vietnamese first-year medical studentsPLOS ONE

Dear Dr. Huang,

Thank you for submitting your manuscript to PLOS ONE. After careful consideration, we feel that it has merit but does not fully meet PLOS ONE’s publication criteria as it currently stands. Therefore, we invite you to submit a revised version of the manuscript that addresses the points raised during the review process.

I agree with the reviewer about the need for a minor revision.

We look forward to receiving your revised manuscript.

Kind regards,

Diego A. Forero, MD; PhD

Academic Editor

PLOS ONE

Journal Requirements:

Reviewers' comments:

Reviewer's Responses to Questions

**Comments to the Author**

1. If the authors have adequately addressed your comments raised in a previous round of review and you feel that this manuscript is now acceptable for publication, you may indicate that here to bypass the “Comments to the Author” section, enter your conflict of interest statement in the “Confidential to Editor” section, and submit your "Accept" recommendation.

Reviewer #1: All comments have been addressed

Reviewer #3: All comments have been addressed

2. Is the manuscript technically sound, and do the data support the conclusions?

Reviewer #1: Yes

Reviewer #3: Yes

3. Has the statistical analysis been performed appropriately and rigorously? 

Reviewer #1: Yes

Reviewer #3: Yes

4. Have the authors made all data underlying the findings in their manuscript fully available?

Reviewer #1: Yes

Reviewer #3: No

5. Is the manuscript presented in an intelligible fashion and written in standard English?

Reviewer #1: Yes

Reviewer #3: Yes

6. Review Comments to the Author

Reviewer #1: Thank you for your work. I really enjoyed reading the updated version of the manuscript. I have one very minor request. Please improve the quality of the figure 1 and 2. I think there are ways to improve the image quality. The written part of the figures are not clearly visible.

Reviewer #3: Thank you, for to opportunity to review this paper about the source of stress and coping modalities amongst first year Medical Students in Vietnam. Utilizing validated measurement instruments, the study identifies key stressors and coping mechanisms prevalent among this demographic. he findings underscore the significance of addressing stress management strategies to support the well-being of medical students.

The authors mentioned that the university has adopted a new reform medical program and that is a more stressful for the students.

P 14

‘’76 The University of Medicine and Pharmacy (UMP) in Ho Chi Minh City has adopted a

77 reformed medical program, which is outcome-based medical education. This reformed

78 program was considered with higher demands and more stressful than the old program,

79 particularly for first-year students who have just entered medical training.’’

Please describe the characteristics of the program, why do you consider it more stressful, what is the difference with the previous program? Do you have any evidence for this? And why the first-year students are affected, did the change was in the middle of university year? Otherwise the current program is the only thing they no directly.

The study was carried in 2020, the students were online or onsite? As it was during the pandemic.

The references are not in Vancouver style as required by the Journal.

Reference 15 missing the Journal and reference 19 either is missing the Journal or it is not a paper and should be quoted differently.

7. PLOS authors have the option to publish the peer review history of their article (what does this mean?). If published, this will include your full peer review and any attached files.

Reviewer #1: No

Reviewer #3: No

---

## [Author Response · Author response to Decision Letter 1]

25 Mar 2024

Reviewer 1

1. I have one very minor request. Please improve the quality of the figure 1 and 2. I think there are ways to improve the image quality. The written part of the figures are not clearly visible.

We have already uploaded the new versions of Figure 1 and Figure 2, using the tool provided in the instruction.

Reviewer 2

1. The authors mentioned that the university has adopted a new reform medical program and that is a more stressful for the students.

P 14

‘’The University of Medicine and Pharmacy (UMP) in Ho Chi Minh City has adopted a reformed medical program, which is outcome-based medical education. This reformed program was considered with higher demands and more stressful than the old program, particularly for first-year students who have just entered medical training.’’

Please describe the characteristics of the program, why do you consider it more stressful, what is the difference with the previous program? Do you have any evidence for this? And why the first-year students are affected, did the change was in the middle of university year? Otherwise the current program is the only thing they no directly.

We have elaborated more details of the reformed program and explain why we perceived it more stressful for medical students, particularly for first-year students. Please refer to Line 77 – Line 86 of the revised manuscript: (We also added the reference of the reformed program of the university from 2016 – reference 18)

This reformed program was considered with higher demands than the old program. Students were introduced to family- and community-based medicine in the early phase of education. New components were integrated in the curriculum, namely Professionalism, Interprofessional education, Scholarly project, and Practice of medicine with the purpose of training students on basic clinical skills in the first two years. A student-centered approach was adopted in all teaching and learning settings 1. The self-directed learning requires students to be more proactive, which is stressful for students who are not familiar with this new education method, particularly the first-year students who have just entered medical training.

2. The study was carried out in 2020, the students were online or onsite? As it was during the pandemic.

During the time we collected these surveys, the students still studied at the university (onsite). The study was conducted before the surge in 2021, when we had a strict quarantine policy.

3. The references are not in Vancouver style as required by the Journal.

Reference 15 missing the Journal and reference 19 either is missing the Journal or it is not a paper and should be quoted differently.

We have revised the references following the Vancouver style and added the Journal of reference 15 and reference 19, which has become reference 20.

---

## [Decision Letter · Decision Letter 2]

16 May 2024

PONE-D-23-15195R2Sources of stress, coping strategies and associated factors among Vietnamese first-year medical studentsPLOS ONE

Dear Dr. Huang,

Thank you for submitting your manuscript to PLOS ONE. After careful consideration, we feel that it has merit but does not fully meet PLOS ONE’s publication criteria as it currently stands. Therefore, we invite you to submit a revised version of the manuscript that addresses the points raised during the review process.

I agree with the reviewer about the need for a further revision of the manuscript.

We look forward to receiving your revised manuscript.

Kind regards,

Diego A. Forero, MD; PhD

Academic Editor

PLOS ONE

Journal Requirements:

Reviewers' comments:

Reviewer's Responses to Questions

**Comments to the Author**

1. If the authors have adequately addressed your comments raised in a previous round of review and you feel that this manuscript is now acceptable for publication, you may indicate that here to bypass the “Comments to the Author” section, enter your conflict of interest statement in the “Confidential to Editor” section, and submit your "Accept" recommendation.

Reviewer #1: All comments have been addressed

Reviewer #4: (No Response)

2. Is the manuscript technically sound, and do the data support the conclusions?

Reviewer #1: Yes

Reviewer #4: No

3. Has the statistical analysis been performed appropriately and rigorously? 

Reviewer #1: Yes

Reviewer #4: No

4. Have the authors made all data underlying the findings in their manuscript fully available?

Reviewer #1: Yes

Reviewer #4: No

5. Is the manuscript presented in an intelligible fashion and written in standard English?

Reviewer #1: Yes

Reviewer #4: Yes

6. Review Comments to the Author

Reviewer #1: (No Response)

Reviewer #4: Thank you for the opportunity to review this paper on stress and coping strategies among medical students in Vietnam. The authors hypothesize that the newly introduced medical education program may contribute to increased stress among medical students compared to the previous program and conducted a survey to investigate the psychosocial backgrounds of medical students. While the results provide some evidence for the need for psychological support for medical students, there are significant concerns that remain for publication as a research paper. In particular, there are serious methodological errors that require substantial revisions before resubmission. Below are the points of concern:

The authors discuss the analysis results of this study, suggesting that the newly introduced outcome-based medical education program may be a contributing factor to stress among medical students. However, as there is no comparison with data from before the new program was introduced, the discussion appears speculative. Please remove statements that may lead to misinterpretation. Additionally, please consider citing studies on stress levels among medical students in other countries and discuss whether stress levels among medical students in Vietnam are particularly high.

In this study, multivariate analysis was conducted using multiple regression analysis. However, the authors' description does not ensure reproducibility. Moreover, only partial results are presented in the tables. Please clearly specify the dependent variables and the independent variables included for each test. It is also not mentioned whether forced entry or stepwise methods were used for independent variable selection.

As per my understanding, it seems that the authors conducted 15 repeated multiple regression analyses, with sociodemographic characteristics (age, sex, father education, mother education, part-time-job, physical stress, psychosocial stress, COVID-19 related stress, stress level) as independent variables, and a total of 15 items from HESI and Brief COPE as dependent variables. This testing method appears to have significant flaws. Since each item of HESI and Brief COPE uses a Likert-type scoring method, they are categorical variables rather than continuous variables, making their use as dependent variables in multiple regression analysis methodologically incorrect. It would be more appropriate to conduct multinomial logistic regression analyses for each item of HESI and Brief COPE to calculate odds ratios. Additionally, correction for multiple testing should be applied. Please conduct a fundamental review of the methodology.

The authors mention the high participation rate of the study participants (97%) as a strength of the study. However, we are concerned that this high participation rate may reflect a distortion of participants' voluntary consent. Please clarify the relationship between the authors and the students. Would you say there is any possibility at all that vulnerable students were coerced into participating in the study? Explain the ethical concerns.

7. PLOS authors have the option to publish the peer review history of their article (what does this mean?). If published, this will include your full peer review and any attached files.

Reviewer #1: No

Reviewer #4: No

---

## [Author Response · Author response to Decision Letter 2]

30 Jun 2024

Dear Reviewers, 

We are grateful for your time and effort reviewing our manuscript. Your comments and suggestions directed us to improve our analysis and writing. Kindly refer to the point-by-point revisions we conducted in response to your comments.

Reviewer 4: 

The authors discuss the analysis results of this study, suggesting that the newly introduced outcome-based medical education program may be a contributing factor to stress among medical students. However, as there is no comparison with data from before the new program was introduced, the discussion appears speculative. Please remove statements that may lead to misinterpretation. Additionally, please consider citing studies on stress levels among medical students in other countries and discuss whether stress levels among medical students in Vietnam are particularly high.

- Thank you for your opinion and recommendations. We have removed the description of the transformed outcome-based medical education program of the university. We also added the statement and references of the situation of stress among medical students in Vietnam (Line 48 – 50 in the revised manuscript) in the Introduction section as following:

“Studies in Vietnam show that the prevalence of experiencing stress among medical students are high, with more than 30% of the students perceived moderate to high level of stress.”

- We added a statement to elaborate the rationale to conduct this study among first year students (Line 80 in the revised manuscript) in the Introduction section as following:

“To date, few studies have directly examined the associations between specific sources of stress and types of coping strategies, particularly in the early phase of medical training.”

In this study, multivariate analysis was conducted using multiple regression analysis. However, the authors' description does not ensure reproducibility. Moreover, only partial results are presented in the tables. Please clearly specify the dependent variables and the independent variables included for each test. It is also not mentioned whether forced entry or stepwise methods were used for independent variable selection.

Thank you for your question to clarify this part. We provided more details about data analysis approaches as your recommendations (Line 203 – 209 in the revised manuscript) in the Analytical approach as following:

“To assess the factors associated with sources of stress and coping strategies, multiple linear regressions, with forced entry regressions, were estimated. For each source of stress as dependent variable, the independent variables included all sociodemographic factors: sex, age, parental educational levels, part-time job, level of perceived stress, and other stress-related variables (acute stress event, physical stress, psychological stress, Covid-19-related stress). Similarly for each coping strategy as dependent variable, we run different regression models that included different sources of stress and all the above sociodemographic and stress-related factors as independent variables.”

As per my understanding, it seems that the authors conducted 15 repeated multiple regression analyses, with sociodemographic characteristics (age, sex, father education, mother education, part-time-job, physical stress, psychosocial stress, COVID-19 related stress, stress level) as independent variables, and a total of 15 items from HESI and Brief COPE as dependent variables. This testing method appears to have significant flaws. Since each item of HESI and Brief COPE uses a Likert-type scoring method, they are categorical variables rather than continuous variables, making their use as dependent variables in multiple regression analysis methodologically incorrect. It would be more appropriate to conduct multinomial logistic regression analyses for each item of HESI and Brief COPE to calculate odds ratios. Additionally, correction for multiple testing should be applied. Please conduct a fundamental review of the methodology.

- Thank you for your recommendation. We have reviewed the analysis and revised the analysis, basically on the types of the independent variables (changed the type of Father educational level, Mother educational level, and Perceived stress levels into Categorical variables instead of Continuous variables as in the previous analysis). We have revised the results in Table 2 and Table 3 in the revised manuscripts, as following:

Table 2: Factors associated with the six sources of stress

Factors Estimates p-value

Mismatch in professional role expectations 

Low perceived stress level -0.32 0.003

Worries about future competence/endurance 

Being male student -0.23 <0.001

Low perceived stress level -0.28 0.033

Financial concerns 

Experiencing acute stress event 0.20 0.027

Low identity of medical profession 

Being male student -0.14 0.007

Having part-time job 0.28 0.005

Non-supportive educational environment 

Low perceived stress level -0.28 0.004

Moderate perceived stress level -0.19 0.030

For each source of stress, we ran a separate regression. The regression models were adjusted for: sex, age, parental educational levels, part-time job, level of perceived stress, acute stress event, physical stress, psychological stress, Covid-19-related stress.

Only factors with statistical significance were listed in Table 2.

Table 3: Factors associated with the nine coping strategies

Factors Estimates p-value

Problem solving 

Low identity of medical profession -0.11 0.029

Social support 

Low identity of medical profession -0.25 <0.001

Avoidance 

Mismatch in professional role expectations 0.22 <0.001

Substance use 

Mismatch in professional role expectations 0.25 <0.001

Non-supportive educational environment 0.15 0.011

Having physical issues 0.15 0.033

Having part-time job 0.20 0.048

Self-blame 

Worries about future competence/ endurance 0.14 0.013

Financial concerns 0.17 <0.001

Academic workload 0.18 0.003

Religion 

Mismatch in professional role expectations 0.30 <0.001

Being male student -0.21 0.013

Humor 

Mismatch in professional role expectations 0.25 <0.001

Being male student 0.19 0.020

Self-distraction 

Having psychological issues -0.23 0.013

For each coping strategy, we ran a separate regression. The regression models were adjusted for: sources of stress, sex, age, parental educational levels, part-time job, level of perceived stress, acute stress event, physical stress, psychological stress, Covid-19-related stress.

Only factors with statistical significance were listed in Table 3.

- We added the elaboration of the findings in Table 2 and Table 3 (Line 245 – 251, Line 258 – 259 and Line 267 – 270 in the revised manuscript) in the Results section as following:

“Students who reported high perceived stress level are more likely to get stressed due to “Mismatch in professional role expectations”, “Worries about future competence/endurance”, and “Non-supportive educational environment”. Students experiencing acute stress event were more likely to have financial concerns compared to other students (β=0.20). On the other hand, “Low identity of medical profession” was found associated with female students and students who had part-time jobs. Female students are also more likely to have “Worries about future competence/endurance”.”

“Table 3 illustrates a variety of associated factors with coping strategies utilized by the participants. Students with “Low identity of medical profession” were less likely to have orientations to solve the problem (β=-0.11) or to seek social support (β=-0.25).”

“Male student tent to adopt humor strategy (β=0.19, p=0.020), while less likely to utilize religious practices (β=-0.21, p=0.013). Meanwhile, students experiencing psychological issues tend to not use “Self-distraction” as a coping orientation to their problems (β= -0.23, p=0.013).”

- Regarding the statistical characteristics of the dependent variables (six sources of stress and nine coping strategies), please refer to the following elaboration:

Although items in the two tools Higher Education Stress Inventory (HESI) and Brief Coping Orientation to Problems Experienced (Brief COPE), were scored using the Likert-type scales, they are typically treated as scores rather than categories, and analyzed based on the summation of scores from clusters of items rather than on the score of an individual item. For HESI, the higher scores indicate higher levels of perceived stressor (1-3). Similarly, for Brief COPE, the higher scores indicate the more frequently the coping orientations were adopted (4-6). In this study, we analyzed the data as previous studies. We conducted analysis to examine the level of perceived stressors, the frequency of adopting coping orientations, and the association (if any) between the perceived stressors and the coping orientations.

Following your recommendations, we’ve performed the binominal logistic regression, in which each factor of HESI and each factor of Brief COPE were treated as binary variable. To be more specific, for HESI, each factor has two categories as Yes (I agree I have this stressor) and No (I disagree I have this stressor). For Brief COPE, each factor has two categories as Yes (I usually adopt this coping orientation to experienced problems) and No (I do not usually adopt this coping orientation to experienced problems).

We have run the Binominal Logistic Regressions and please refer to the findings in Appendix enclosed to this Response, but we have opted to present the data in the same manner as previous studies (1-6).

The authors mention the high participation rate of the study participants (97%) as a strength of the study. However, we are concerned that this high participation rate may reflect a distortion of participants' voluntary consent. Please clarify the relationship between the authors and the students. Would you say there is any possibility at all that vulnerable students were coerced into participating in the study? Explain the ethical concerns.

This study is among the three studies of a project to provide the Transforming Stress Program to all first-year students. This Transforming Stress Program was introduced to all first-year students in their orientation session at the beginning of the school year. It was introduced as an extra-curricular activity, and there was no other measure taken to enhance participation. The fact that it was endorsed by the school administration may help explain the high participation rate. Before they took the training, we invited them to fill out the survey to have the baseline information about their current stress mindset, stressors, and coping strategies on a voluntary basis. The process involves nothing of a coercive nature.

The Principal Investigator was an alumnus of the University and has no relationship with the first-year students. This study had been approved by the IRB of the University. We introduced the study to all students and got their agreement to join in this study by the written Informed Consent Forms.

1. Saxena SK, Mani RN, Dwivedi AK, Ryali V, Timothy A. Association of educational stress with depression, anxiety, and substance use among medical and engineering undergraduates in India. Industrial psychiatry journal. 2019;28(2):160-9.

2. Pacheco JPG, Hoffmann MS, Braun LE, Medeiros IP, Casarotto D, Hauck S, et al. Translation, cultural adaptation, and validation of the Brazilian Portuguese version of the Higher Education Stress Inventory (HESI-Br). Trends in psychiatry and psychotherapy. 2023;45:e20210300.

3. Shim E-J, Jeon HJ, Kim H, Lee K-M, Jung D, Noh H-L, et al. Measuring stress in medical education: validation of the Korean version of the higher education stress inventory with medical students. BMC Medical Education. 2016;16.

4. Matsumoto S, Yamaoka K, Nguyen HDT, Nguyen DT, Nagai M, Tanuma J, et al. Validation of the Brief Coping Orientation to Problem Experienced (Brief COPE) inventory in people living with HIV/AIDS in Vietnam. Global health & medicine. 2020;2(6):374-83.

5. García FE, Barraza-Peña CG, Wlodarczyk A, Alvear-Carrasco M, Reyes-Reyes A. Psychometric properties of the Brief-COPE for the evaluation of coping strategies in the Chilean population. Psicologia: Reflexão e Crítica. 2018;31(1):22.

6. Babakhani M, Aghabarary M, Norouzinia R. Perceived stress and coping strategies after unsuccessful cardiopulmonary resuscitation among pre-hospital emergency technicians: A multicenter cross-sectional study. Heliyon. 2024;10(10):e31418.

---

## [Decision Letter · Decision Letter 3]

11 Jul 2024

Sources of stress, coping strategies and associated factors among Vietnamese first-year medical students

PONE-D-23-15195R3

Dear Dr. Huang,

We’re pleased to inform you that your manuscript has been judged scientifically suitable for publication and will be formally accepted for publication once it meets all outstanding technical requirements.

Kind regards,

Diego A. Forero, MD; PhD

Academic Editor

PLOS ONE

Additional Editor Comments (optional):

Reviewers' comments:

Reviewer's Responses to Questions

**Comments to the Author**

1. If the authors have adequately addressed your comments raised in a previous round of review and you feel that this manuscript is now acceptable for publication, you may indicate that here to bypass the “Comments to the Author” section, enter your conflict of interest statement in the “Confidential to Editor” section, and submit your "Accept" recommendation.

Reviewer #4: All comments have been addressed

2. Is the manuscript technically sound, and do the data support the conclusions?

Reviewer #4: Yes

3. Has the statistical analysis been performed appropriately and rigorously? 

Reviewer #4: Yes

4. Have the authors made all data underlying the findings in their manuscript fully available?

Reviewer #4: Yes

5. Is the manuscript presented in an intelligible fashion and written in standard English?

Reviewer #4: Yes

6. Review Comments to the Author

Reviewer #4: The authors responded to my comments and the quality of the paper has improved in terms of statistical analysis and discussion.

7. PLOS authors have the option to publish the peer review history of their article (what does this mean?). If published, this will include your full peer review and any attached files.

Reviewer #4: No

---

## [Editor Report · Acceptance letter]

22 Jul 2024

PONE-D-23-15195R3 

PLOS ONE

Dear Dr. Huang, 

I'm pleased to inform you that your manuscript has been deemed suitable for publication in PLOS ONE. Congratulations! Your manuscript is now being handed over to our production team.

Kind regards, 

on behalf of

Dr. Diego A. Forero 

Academic Editor

PLOS ONE